# ^1^H NMR-Based Metabolic Profiling to Follow Changes in Pomelo Cultivars during Postharvest Senescence

**DOI:** 10.3390/foods12102001

**Published:** 2023-05-15

**Authors:** Juan Liu, Xinqiao Zhou, Dagang Chen, Jie Guo, Ke Chen, Chanjuan Ye, Chuanguang Liu

**Affiliations:** Rice Research Institute, Guangdong Academy of Agricultural Sciences, Guangdong Rice Engineering Laboratory, Guangdong Key Laboratory of New Technology in Rice Breeding, Key Laboratory of Genetics and Breeding of High Quality Rice in Southern China (Co-Construction by Ministry and Province), Ministry of Agriculture and Rural Affairs, Guangdong Rice Engineering Laboratory, Guangzhou 510640, China

**Keywords:** pomelo, metabolic profiling, NMR, naringin, citric acid, flavor

## Abstract

This study investigated metabolite changes in three pomelo cultivars during postharvest senescence using ^1^H NMR-based metabolic profiling. Three pomelo cultivars, ‘Hongroumiyou’, ‘Bairoumiyou’ and ‘Huangroumiyou’, abbreviated as “R”, “W” and “Y” according to the color of their juice sacs, were stored at 25 °C for 90 days, and NMR was applied to determine the metabolite changes in juice sacs during storage. Fifteen metabolites were identified, including organic acids, sugars, amino acids, fatty acids, phenols and naringin. Partial least squares discriminant analysis (PLS-DA) was used to screen the significant metabolites according to the variable importance for the projection (VIP) scores in three pomelo cultivars during 90 days of storage. Additionally, eight metabolites, naringin, alanine, asparagine, choline, citric acid, malic acid, phosphocholine and β-D-glucose, were screened to be the crucial biomarkers with VIP > 1. The undesirable flavor of “bitter and sour” during the 60 days of storage was mainly attributed to the naringin, citric acid and sugars. According to the correlation analysis, the citric acid content determined by NMR showed a significantly positive relationship with that analyzed by HPLC. These findings suggested that NMR technology was accurate and efficient for metabolomic analysis of pomelo fruit, and the ^1^H NMR-based metabolic profiling can be efficient during quality evaluation and useful for improving the fruit flavor quality during postharvest storage.

## 1. Introduction

Pomelo (*Citrus maxima* (Burm.) Merr.), which is widely cultivated and broadly consumed for its nutrition and flavor, belongs to non-climacteric citrus fruit; thus, no respiration peak would be observed during postharvest storage. However, granulation would occur in juice sacs and softening could appear in pericarp of pomelo fruit during postharvest storage, which is mainly attributed to lignin accumulation and water loss that could result in poor taste and reduced marketability [1]. It is quite well documented that the granulation process could probably be attributed to the gelation occurrence induced by the free water fixation caused by pectin remodeling in juice sacs [2]. Our previous work revealed that lignin accumulation was closely associated with energy metabolism, lipid peroxidation and sugar metabolism [3,4]. In addition to lignification, softening is also significant in pomelo fruit, which was reinforced by the observation that antioxidant and cell wall metabolisms could result in softening and senescence of pummelo fruit, as described in our previous work [5]. In addition to granulation, lignification and softening, another undesirable trait in postharvest pomelo fruit is the variable taste and flavor. As a major organic acid in citrus fruit, citric acid is the decisive factor of fruit sour taste and acts as the primary substrate of respiration metabolism. Citric acid degradation occurs during fruit ripening [6,7,8]; however, it is still unclear how citric acid changes and what metabolites affect the flavor of pomelo fruit during postharvest storage. To better elucidate the relationship between flavor and metabolites, we conducted a study on the changes in metabolites of different varieties of pomelo fruit during storage to explore the key metabolites that affect the flavor and quality changes of pomelo fruit.

Among the current analytical technologies, ^1^H NMR spectroscopy is an increasingly trustworthy tool that has provided quantitative and qualitative high-throughput data for food metabolomics analysis in recent years [9,10]. For the advantages of speed and simplicity of samples preparation, information on primary and secondary metabolites can be obtained simultaneously, with high reproducibility and non-destruction. ^1^H-NMR has also been widely recognized as an effective tool to display its promising potential in metabolites analysis of foods and beverages [11,12,13,14]. Subsequently, the NMR data are frequently processed by multivariate data analysis, such as principal component analysis (PCA), and partial least squares discriminant analysis (PLS-DA) has usually been utilized to screen the crucial biomarkers between samples [15,16]. By applying this approach, extensive studies have been successfully conducted to investigate discrimination of industrial and craft beers [17], raw and cooked pufferfish (*Takifugu flavidus*) meat [18], wild tropical tunas by species, size category and geographic origin [14], free-range and barn-raised broiler chickens [19] and different cultivars of green tea (*Camellia sinensis*) [12]. Our previous work conducted in okra indicated that NMR spectroscopy was a promising tool for identifying the key metabolites that determine okra quality during postharvest storage [15]. Therefore, NMR technology might as well play an unexpected role in revealing the metabolites that affect the flavor and quality changes during postharvest pomelo storage.

In this work, NMR technology was applied to determine metabolites changes in three pomelo cultivars during 90 days of storage. PLS-DA model was used to analyze the NMR data and screen the significant metabolites to influence the flavor and taste of three pomelo cultivars during postharvest senescence. Additionally, the correlation analysis between the peak area of citric acid collected by NMR technology and actual content analyzed by HPLC was conducted to verify the accuracy of NMR technology in determining the metabolites changes in pomelo fruit.

## 2. Materials and Methods

### 2.1. Plant Materials and Treatment

Three pomelo cultivars (“Hongroumiyou”, “Bairoumiyou” and “Huangroumiyou”) were harvested at commercial maturity from a local orchard in Dapu county, Meizhou city, Guangdong province, China. Forty fruits from each cultivar were selected and divided into four groups. Each fruit was packed in a polyethylene bag (0.03 mm thickness) and stored at 25 °C and 70–80% relative humidity for 0, 30, 60 and 90 days (d). Juice sacs from ten fruits at each storage period was sampled and ground into powder under liquid nitrogen for analysis.

### 2.2. Metabolites Detection by NMR Spectroscopy

Fifty milligrams of juice sacs from ten fruits at each storage time with three replicates were sampled for NMR analysis according to our previously reported methods [15]. Briefly, the pomelo powder was immersed in methanol-d4 for 1 min and then extracted via ultrasound for 1 h at 40 °C. After centrifugation (15,000× *g*, 10 min), the upper fraction was subjected to NMR analysis. The experimental conditions of ^1^H NMR and 2D NMR spectra were conducted according to our previous methods. ^1^H NMR spectra were recorded at 500.13 MHz proton frequency on a BRUKER AVANCE III 500 spectrometer (Bruker, Karlsruhe, Germany) equipped with a TCI cryoprobe and Z-gradient system at 297 K. For signal assignment purposes, 2D NMR spectra were acquired at 25 °C, including ^1^H-^1^H correlation spectroscopy (COSY), selective total correlation spectroscopy (1D-TOCSY), J-resolved spectroscopy (JRES), ^1^H-^13^C heteronuclear single quantum coherence spectroscopy (HSQC) and heteronuclear multiple bond correlation (HMBC). For COSY and 1D-TOCSY experiments, 128 transients were collected into 1024 data points for each of the 160 increments, with a spectral width of 12.0 ppm for both dimensions. A phase-insensitive mode with gradient selection was used for the COSY experiments and MLEV-17 was employed as the spin-lock scheme in the phase-sensitive 1D-TOCSY experiment with a mixing time of 80 ms. For JRES spectra, 128 transients were collected into 4096 data points for each of the 80 increments with a spectral width of 6000 Hz in the acquisition and 60 Hz in the evolution dimensions. ^1^H-^13^C HSQC and HMBC experiments were recorded by using the gradient selected sequences with 512 transients and 2048 data points for each of the 128 increments. The spectral widths were set at 6000 Hz for ^1^H, 20,625 Hz for ^13^C in HSQC experiments and 27,500 Hz for ^13^C HMBC experiments, respectively. The data were zero-filled to a 2000 × 2000 matrix with appropriate window functions prior to Fourier transformation.

### 2.3. Determination of Citric Acid

The measurement and extraction of organic acids were modified from a method put forward by previous reports [20]. Briefly, juice sacs from three pomelo cultivars were frozen in liquid nitrogen and ground into powder. A total of 3 g of powdered sample was weighed and extracted with 3 mL deionized water via vortexing for 2 min and followed by ultrasonic extraction at 40 °C for 1 h. The extract was centrifuged at 12,000× *g* for 10 min and the supernatant was subjected to HPLC analysis. Extracts were analyzed by ALTUSA-10 consisting of a differential refraction detector, equipped with C18 column (5 μm, 4.6 × 250 mm). The column temperature was 40 °C. The injection volume was 10 μL. Absorbance was measured at 210 nm. Additionally, 0.1% phosphoric acid solution (solvent A) and methanol (solvent B) were used as mobile phases at a flow rate of 1 mL min^−1^. The linear gradient elution was performed as follows: 0–4 min, 95% A; 4–5 min, 95% − 5% A; 5–10 min, 5% A; 10–15 min, 95% A. Identification of citric acid was carried out by comparing the retention time and UV absorption of authentic standards.

### 2.4. Statistical Analysis

The results were expressed as means ± standard errors (SE) (*n* = 3). Significant differences between two groups examined using the software SPSS (version 22.0, IBM, New York, NY, USA) via ANOVA (analysis of variance) using Duncan’s test with least significant differences at 0.05 level.

### 2.5. NMR Data Processing and Multivariate Data Analysis

^1^H NMR spectra of pomelo extract was manually corrected for phase and baseline distortions and referenced to MeOD-*d*_4_ (3.31 ppm) by using the software package TOPSPIN (v3.2, Bruker Biospin, Germany). Fifteen metabolites were identified by 1D and 2D NMR spectra. The relative contents of these compounds were defined by the peak area. A linear baseline scaling normalization approach was used. The baseline was constructed by calculating the median of each feature over all spectra. The scaling factor was computed for each spectrum as the ratio of the mean intensity of the baseline to the mean intensity of the spectrum. The intensities of all spectra were multiplied by their particular scaling factors.

The NMR data of metabolites were analyzed for partial least squares discriminant analysis (PLS-DA) using SIMCA 15 (Umetrics, Malmo, Sweden). The external variable “time” was used as quantitative and the external variable “cultivar” was used as qualitative. The predictive variable importance for the projection (VIP) was used for principal metabolites analysis and VIP > 1 indicate ‘crucial’ variables. Hence, we mainly focused on those metabolites with VIP > 1 in the following analysis.

## 3. Results

### 3.1. Identification of Metabolites in Pomelo Extract by 1D and 2D NMR

The ^1^H NMR spectra was shown in Figure 1. Fifteen metabolites were identified, including fatty acids, amino acids, organic acids and phenolics. Proton signals of the metabolites were listed in Table 1. ^1^H NMR spectra of extract from R, W and Y was shown in Figure 2, and 2D NMR spectra of pomelo extract was shown in Figure 3.

Signals in the region between 3.00 and 5.50 ppm, which were assigned to the sugar compounds, frequently overlapped, and the major differences were observed in the anomeric signals of sugar compounds. We were able to assign the anomeric protons of sucrose at 5.38 ppm (d, *J* = 3.8 Hz), α-D-glucose at 5.10 ppm (d, *J* = 3.72 Hz), β-D-glucose at 4.46 ppm (d, *J* = 8.0 Hz) and fructose at 4.09 ppm (d, *J* = 8.3 Hz). These compounds were the major compounds in Citrus maxima (Burm.) Merr. The residual proton signals of the sugars shown in the crowded region (3.0–4.0 ppm) were assigned by the comparison of ^1^H NMR spectra of the reference compounds and JRES, COSY, HSQC and HMBC spectra (Figure 3A–D).

The ^1^H NMR spectra in the aromatic region (6.8–8.0 ppm) revealed the presence of phenolics [21]. Six aromatic protons at 6.15 (d, 1H, *J* = 2.20 Hz), 6.17 (d, 1H, *J* = 2.20 Hz), 6.81 (d, 2H, *J* = 8.20 Hz), 7.1 (d, 2H, *J* = 8.20 Hz) indicated the existence of an important secondary metabolite. Further analysis of 1H-1H COSY spectra led to identification of the benzene ring with AX spin and AB system. The signals at 2.76 ppm and 3.16 ppm were assigned to H-1 and H-2. In the HSQC experiments, H-6 (6.15 ppm) correlated with C-6 (96.3 ppm), H-8 (6.17 ppm) correlated with C-8 (95.4 ppm), H-2# and H-6# (7.1 ppm) correlated with C-2# and C-6# (127.6 ppm) and H-3# and H-5# (6.81 ppm) correlated with C-3# and C-5# (114.9 ppm). Thus, these three protons were assigned to naringin.

The signals at 2.55 (dd, *J* = 7.7, 16.2 Hz), 2.79 (m) and 4.34 (m) were assigned to malic acid. The signals at 2.73 (d, *J* = 15.4 Hz) and 2.81 (d, *J* = 15.4 Hz) were assigned to citric acid. The signals at 1.45 (d, *J* = 7.2 Hz) and 3.61 (m) were assigned to alanine. Some other components, including γ-amino-butyrate (GABA) (1.89, 2.40, 2.99 ppm), fatty acids (0.98, 1.24–1.38, 1.55–1.65, 2.32 ppm), choline (3.20 ppm), phosphocholine (3.22 ppm), unsaturated lipids (5.28–5.33, 2.74–2.80 ppm), asparagine (2.69, 2.93, 3.82 ppm), were also assigned in the spectra (Table 1).

### 3.2. Multivariate Data Analysis

In order to find the principal metabolites, the PLS-DA model was employed to determine the critical metabolites variation in three pomelo cultivars during postharvest storage. PLS-DA can establish the relationship model between the expression of metabolites and the sample category to realize the prediction of the sample category. Variable Importance for the Projection (VIP) can be calculated to measure the influence intensity and interpretation ability of the expression mode of each metabolite on the classification and discrimination of each group of samples, to assist in the screening of marker metabolites (usually with VIP >1 as the screening standard). The score plot shown in Figure 4A,B displayed the identified metabolites ranked based on their VIP scores. According to the score plot of PLS-DA (Figure 4A), the first two components (t [1], t [2]) of PLS-DA accounted for 78.9% of the total variance among the samples. The column with red color was the metabolites with VIP > 1, and those columns with green color represented metabolites with VIP < 1. We took VIP > 1 as the criteria to screen the crucial biomarkers among three pomelo cultivars during 90 days of storage. Eight metabolites, naringin, alanine, asparagine, choline, citric acid, malic acid, phosphocholine, and β-D-glucose, were screened to be the crucial biomarkers.

The peak areas of the above eight metabolites in three pomelo cultivars during 90 days of storage were calculated, as compared in Figure 5. Compared with other metabolites, the content of citric acid is the highest in the same cultivar during 90 days of storage, followed by β-D-Glucose. In the three pomelo varieties, the content of citric acid gradually increased with the extension of storage time, reached the highest level after 60 days of storage and then decreased. The content of β-D-Glucose gradually increased in cultivar R, reached its highest at 60 days, and then decreased. In cultivar “W”, the content of β-D-Glucose reached the highest level after 30 days of storage, and then decreased gradually with the extension of storage time. In cultivar “Y”, The content of β-D-Glucose decreased slightly at first, then rose to the peak at 60 days of storage, and then decreased.

As shown in Figure 5, the naringin content in “R”, “W” and “Y” increased gradually to the maximum from 0 to 60 days, then decreased from 60 to 90 days. However, the naringin content in cultivar “W” showed a slight drop from 0 to 30 days, followed by a slow increase to the maximum from 30 to 60 days, and then decreased from 60 to 90 days. Alanine content increased gradually to the maximum from 0 to 60 days, and then decreased rapidly from 60 to 90 days of storage in cultivar “R”, while that in cultivar “W” kept an increasing tendency during 0 to 90 days. Furthermore, the alanine content in cultivar “Y” decreased from 0 to 30 days, then rapidly increased to the maximum from 30 to 60 days and finally showed a slight drop from 60 to 90 days. The choline content in cultivar “R” and “W” increased from 0 to 30 days, and kept steady from 30 to 60 days, and then declined from 60 to 90 days. In cultivar “Y”, the choline content increased gradually from 0 to 60 days, and then decreased rapidly from 60 to 90 days. The malic acid content in three pomelo cultivars from 0 to 60 days increased gradually to the maximum, kept steady in cultivar “R” but decreased from 60 to 90 days during 60 to 90 days in cultivar “W” and “Y”. The phosphocholine content in cultivar “R” increased to the maximum from 0 to 60 days, and remained almost unchanged from 60 to 90 days, while that in cultivar “W” increased to the maximum from 0 to 30 days, but showed a gradual decreasing trend from 30 to 90 days. Moreover, in cultivar “Y”, the phosphocholine content exhibited a increasing trend from 0 to 60 days, but declined sharply from 60 to 90 days.

### 3.3. Changes in Citric Acid Content in Three Cultivars during Postharvest Storage

As shown in Figure 6, the citric acid content displayed a gradual increase from 0 to 60 days followed by a rapid drop from 60 to 90 days in three pomelo cultivars. The citric acid content in “Y” was higher than “R” during the whole storage time. Compared to the other two cultivars, the citric acid content in “W” was the lowest during the whole storage period. In addition, different citric acid accumulation rules would be observed in different citrus cultivars during fruit senescence. In our work, the citric acid levels in three pomelo cultivars were distinctly different. In cultivar “Y”, the citric acid content showed a sharp increase from 0 to 30 days of storage, followed by a slower increase from 30 to 60 days of storage. However, a rapid decrease was observed during 60 to 90 days storage. As to cultivar “R”, the citric acid from 0 to 60 days showed a gradual increase followed by a gentle drop during 60 to 90 days of storage. The citric acid level in cultivar “W” from 0 to 60 days displayed a slight increase from 0 to 30 days, and then increased faster from 30 to 60 days, followed by a rapid drop during 60 to 90 days of storage.

We chose citric acid with the maximum peak area to verify the accuracy of NMR technology in analyzing the metabolites of pomelo fruit. Hence, we determined the contents of citric acid in three pomelo cultivars and correlation analyses were conducted between the peak area identified by NMR analysis and the actual content of citric acid in three pomelo cultivars during 90 days of storage. Correlation analyses demonstrated that the peak area identified by NMR analysis showed positive relations with the actual contents in cultivar “R”, “W” and “Y” (r = 0.925, *p* < 0.05; r = 0.938, *p* < 0.05; r = 0.959, *p* < 0.05, respectively). These results suggested that the identification of metabolite changes in three pomelo cultivars by application of NMR technology was accurate and efficient.

## 4. Discussion

### 4.1. Metabolite Biomarkers Differentiating Pomelo Cultivars and Storage Times

Phenols are more particularly involved in the antioxidant metabolism and play a key role in scavenging the reactive oxygen species (ROS) during postharvest fruit senescence. Previous studies conducted in longan fruit indicated that the pericarp browning of harvested longan fruit was closely related to phenolic metabolism, and it was found that higher levels of total phenol content may account for lower pulp breakdown of longan pulp [22]. Naringin belongs to the class of flavonoids and exists extensively in citrus fruits. Naringin is a bitter-taste flavanone glycoside found with the molecular weight of 580.4 g/mol, with the molecular formula C_27_H_32_O_14_ in grapes and citrus fruits. It is well documented that naringin exhibits good antifungal, antimycotic, antiobesity, antiulcer, antiaging, anticancer, antioxidant and anti-inflammatory properties [23,24,25,26].

Organic acids contribute greatly to the total acidity of the fruit and play a major role in determining the taste, flavor and aroma of citrus fruit during postharvest storage [27]. Citric acid and malic acid are two major organic acids in most fruits during the ripening process, while citric acid is found to be the leading organic acid in juice sacs of citrus fruit. Citric acid is produced in the tricarboxylic acid (TCA) cycle, which can also be called the citric acid cycle in mitochondria and is the most important intermediate product in the TCA cycle, acting as the metabolism center and the main energy source for organisms. It has been reported that the blocking activity of aconitase in mitochondria could contribute to citric acid accumulation in citrus fruits [7]. The organic acid level was closely related to fruit ripening and senescence and showed a reverse correlation with weight loss of harvested fruit during senescence [28]. Pomelo fruit is abundant in organic acids, of which citric acid and malic acid are two most distinctive ones that determine fruit acidity. The organic acids of forty-seven orange cultivars were compared and citric acid was observed to be the main component in the high-acid cultivars, while malic acid was the key acid in low-acid component cultivars [29]. The citric acid decline in the later storage of pomelo fruit in this work was in accordance with previous studies, which have reported that the total acidity in mandarin, orange and grapefruit showed a tendency to decrease during the maturation process [30,31,32].

Sugars play a fundamental role in plant growth. Sugars are synthesized through photosynthesis and represent mobile and accessible materials available for metabolism, providing energy and a source of carbon skeletons for plant growth and development. Numerous studies have proved that sugars could be involved more particularly in signaling transduction, osmotic homeostasis and response to abiotic stress [33,34]. With regard to fruits, the possible link between the accumulation and proportion of sugars and quality and attractiveness was reinforced by the observation in a recent study that hexose accumulation plays a crucial role in determining the sweetness of mature fruits among citrus cultivars [34]. In addition, sugars play an important role in determining the flavor and taste of fruit, and the sweetness is mainly attributed to the accumulation of total soluble sugars. The critical role of soluble sugars has been highlighted in banana and pomelo fruit in our previous work, and higher total soluble sugars was found to be beneficial to alleviate banana chilling injury symptoms, while higher glucose and fructose may account for the lignin accumulation in pomelo fruit [4,20,21,35]. Glucose is the energy source and the intermediate product of metabolism of living organisms. The β-D-glucose content in the late storage of pomelo fruit decreased, probably in relation to the energy deficit, which was a symbol of senescence and could accelerate the process of pomelo senescence [4].

Two kinds of amino acids, alanine and asparagine, were found to be crucial biomarkers with VIP > 1 in three pomelo cultivars during 90 days of storage and they gave strong credence to the participation of these amino acids in the senescence process of pomelo fruit. Alanine has been reported to be critical for connecting carbon and nitrogen metabolisms and plays a role in the carbon–nitrogen balance in plants [36,37]. Asparagine is the most important amino acid in many plants and has been found to play an important role in storing and transporting nitrogen in living organisms [38]. GABA has been investigated and was found to be closely associated with plants’ anti-adversity ability against environmental stress [39]. A previous study on blueberries found that GABA treatment could better maintain the quality of postharvest blueberries by improving their antioxidant capacity [40].

Fatty acids include unsaturated fatty acids (USFAs) and saturated fatty acids (SFAs) are the main components of membrane lipids, the improper ratio of which could reflect the integrity of the cell membrane [41]. Choline and phosphocholine are important ingredients of the biological membrane and play a critical role in maintaining normal activities of the membrane [42]. The contents of choline and phosphocholine in pomelo fruit changed significantly during the 90 days of storage, revealing that the structure and function of membrane may change greatly during the pomelo senescence process.

### 4.2. A Hypothesis of Flavor Changes in Pomelos during Postharvest Storage

In our previous work, we found that the total soluble sugar content in both cultivar “Y” and “W” displayed a decrease from 0 to 30 days of storage, followed by an increase from 30 to 60 days, and subsequently showed a decrease from 60 to 90 days of storage. Furthermore, in cultivar “R”, the total soluble sugar content increased sharply from 0 d to 30 d of storage, followed by a slower increase; after that, a slight decrease was observed. In this work, during the 60 days of storage, the content of β-D-glucose increased to its peak at 60 d in “R” and “Y”, but that in “W” rose to the highest at 30 d and slightly decreased from 30 d to 60 d. At the same time, the content of naringin that possesses the bitter taste increased, as the level of citric acid maintained an increasing tendency. Both factors resulted in the appearance of a bitter and sour taste in the pomelo fruit (Figure 7). After that, during the 60 days to 90 days of storage, since complex changes occur in metabolites, further investigations will be needed. In a word, the flavor and taste changes during 90 days of storage, and the undesirable flavor of “bitter and sour” during the 60 days of storage was mainly attributed to the naringin and citric acid. In addition, the sugars accumulation may occur; during that process, further investigation will be made.

## 5. Conclusions

In brief, analysis of metabolites profiles performed using ^1^H high-resolution NMR is a promising tool to illuminate the metabolites changes in three pomelo cultivars during 90 days of storage. Fifteen metabolites were identified during 1D and 2D NMR experiments, highlighting the presence of organic acids, sugars, amino acids, fatty acids, phenols and naringin. PLS-DA allowed discrimination of pomelo, in which naringin, alanine, asparagine, choline, citric acid, malic acid, phosphocholine, β-D-glucose were found to be involved the most, particularly in three pomelo cultivars during the senescence process. The citric acid declined from 60 to 90 days, as shown in the cultivars “Y”, “R” and “W”, descending from high to low during postharvest storage. The ^1^H NMR-based metabolic profiling is promising in rapid quality evaluation during different species and contributes to the continuous improvement of quality during storage and processing. We chose a semi-quantitative method of NMR-based metabolomics, which involves directly dissolving the sample with methanol-d4 and conducting a comprehensive preliminary analysis of the metabolites. However, only a small fraction of the metabolites have been revealed and a more extensive assignment with more detected metabolites is expected. Therefore, we will further use NMR spectroscopy as a quantitative method to analyze the metabolites changes in the following study, since NMR spectroscopy is known to be a quantitative method if the full relaxation condition is respected. It would have been sufficient to add a known amount of the reference substance (visible in the spectra at 0.0 ppm) to calibrate the concentrations of each metabolite.

## Figures and Tables

**Figure 1 foods-12-02001-f001:**
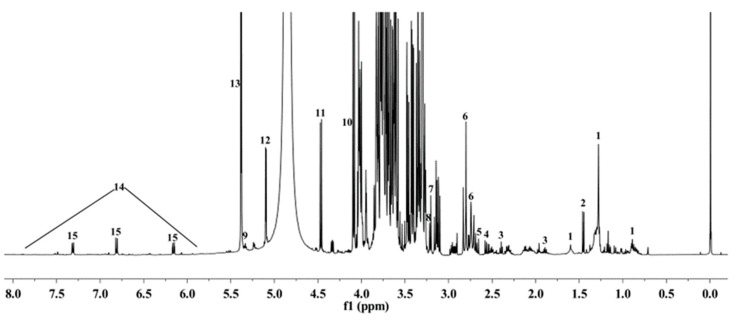
Representative ^1^H NMR spectra of pomelo extract of “Hongroumiyou” (R). Peaks: 1—Fatty acids; 2—Alanine; 3—γ-amino-butyrate (GABA); 4—Malic acid; 5—Asparagine; 6—Citric acid; 7—Choline; 8—Phosphocholine; 9—Unsaturated lipids; 10—Fructose; 11—β-D-Glucose; 12—α-D-Glucose; 13—Sucrose; 14—Phenolics; 15—Naringin.

**Figure 2 foods-12-02001-f002:**
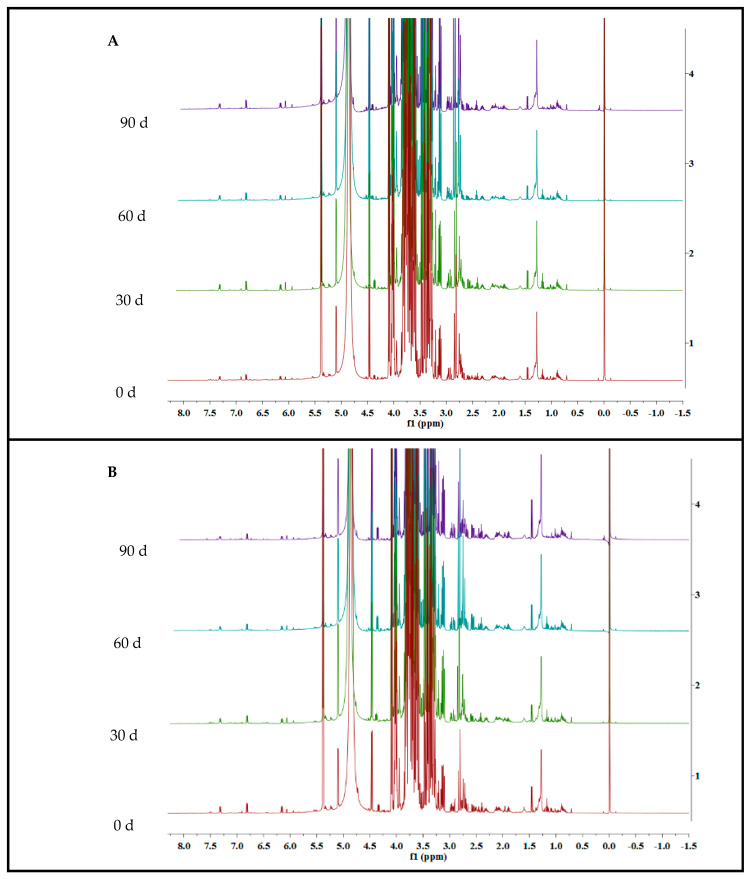
(**A**–**C**) ^1^H NMR fingerprint of pomelo extract from cultivar “Hongroumiyou” (R), “Bairoumiyou” (W) and “Huangroumiyou” (Y) during 90 days of storage. Red, green, blue, and purple represent the ^1^H NMR spectra of the sample at 0, 30, 60, and 90 days, respectively.

**Figure 3 foods-12-02001-f003:**
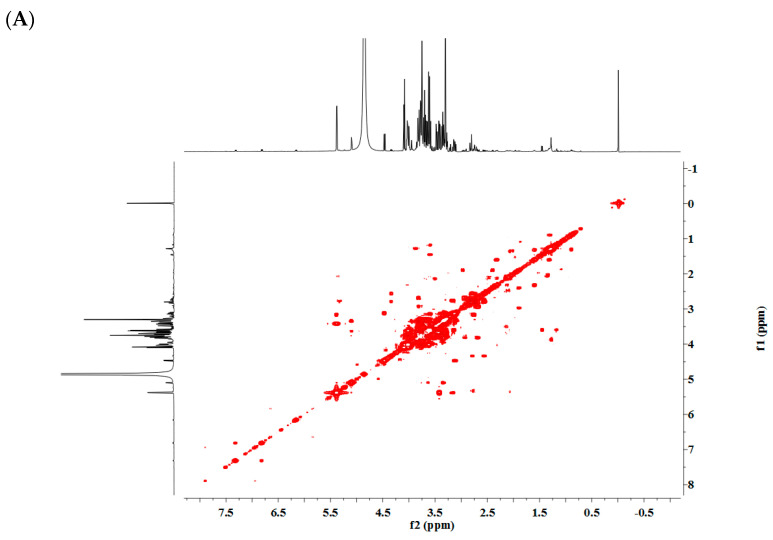
Two-dimensional NMR spectra of pomelo extract. (**A**) COSY spectra; (**B**) JERS spectra; (**C**) HSQC spectra; (**D**) HMBC spectra.

**Figure 4 foods-12-02001-f004:**
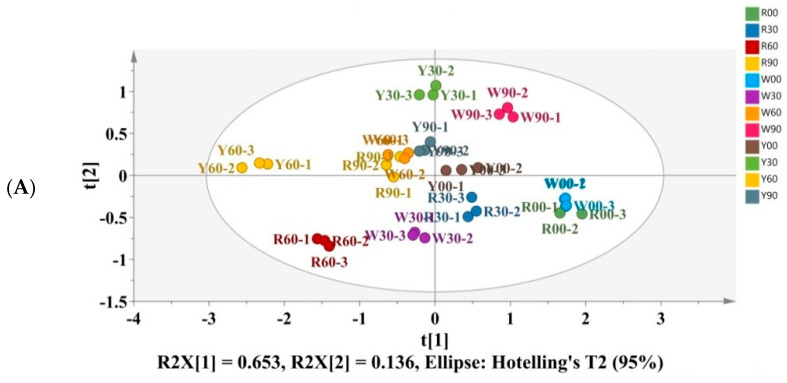
Partial least squares discriminant analysis (PLS-DA) of pomelo metabolites showing t [1] (65.3%) versus t [2] (13.6%). (**A**) Score plot. (**B**) Identified metabolites ranked by VIP scores. (R00-1, R00-2 and R00-3 are “Hongroumiyou” (R) at 0 d of storage; R30-1, R30-2 and R30-3 are “Hongroumiyou” (R) at 30 d of storage; R60-1, R60-2 and R60-3 are “Hongroumiyou” (R) at 60 d of storage; R90-1, R90-2 and R90-3 are “Hongroumiyou” (R) at 90 d of storage. W00-1, W00-2 and W00-3 are “Bairoumiyou” (W) at 0 d of storage; W30-1, W30-2 and W30-3 are “Bairoumiyou” (W) at 30 d of storage; W60-1, W60-2 and W60-3 are “Bairoumiyou” (W) at 60 d of storage; W90-1, W90-2 and W90-3 are “Bairoumiyou” (W) at 90 d of storage. Y00-1, Y00-2 and Y00-3 are “Huangroumiyou” (Y) at 0 d of storage; Y30-1, Y 0-2 and Y30-3 are “Huangroumiyou” (Y) at 30 d of storage; Y60-1, Y60-2 and Y60-3 are “Huangroumiyou” (Y) at 60 d of storage; Y90-1, Y90-2 and Y90-3 are “Huangroumiyou” (Y) at 90 d of storage).

**Figure 5 foods-12-02001-f005:**
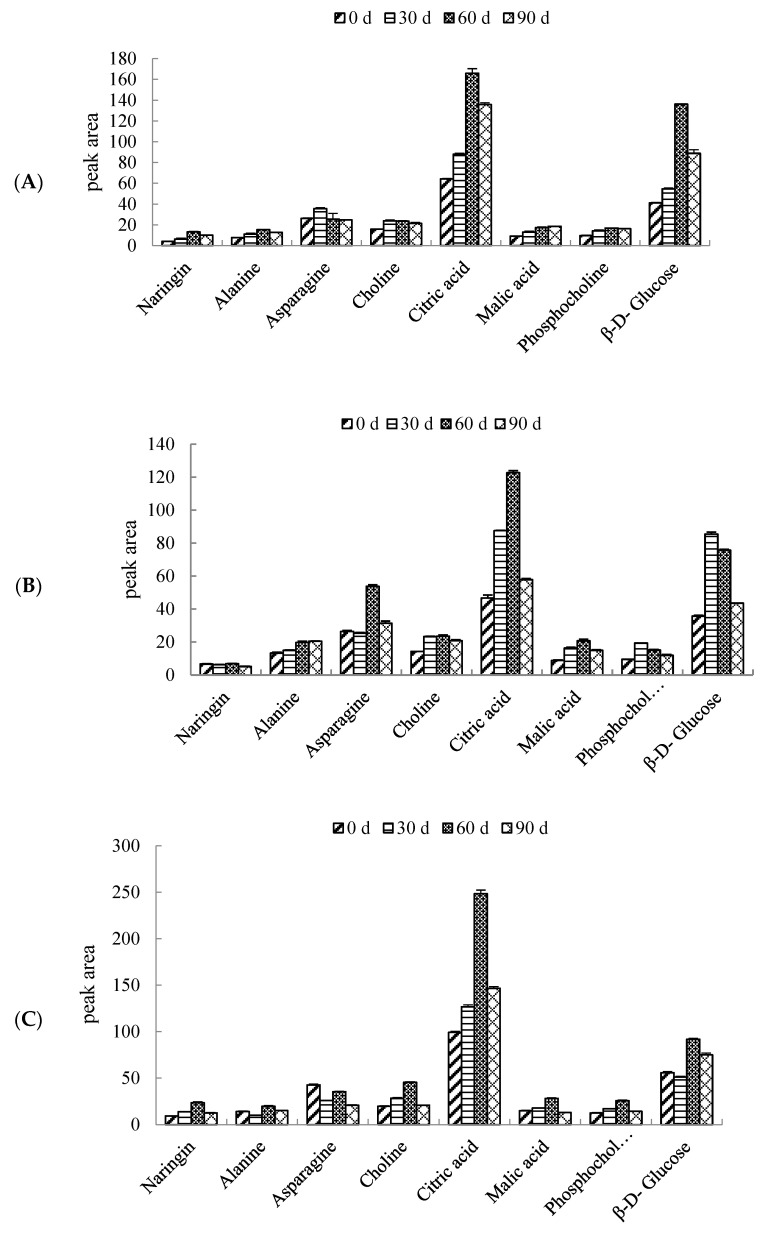
(**A**–**C**) changes in metabolites with VIP > 1 of “Hongroumiyou” (R), “Bairoumiyou” (W) and “Huangroumiyou” (Y) during 90 days of storage.

**Figure 6 foods-12-02001-f006:**
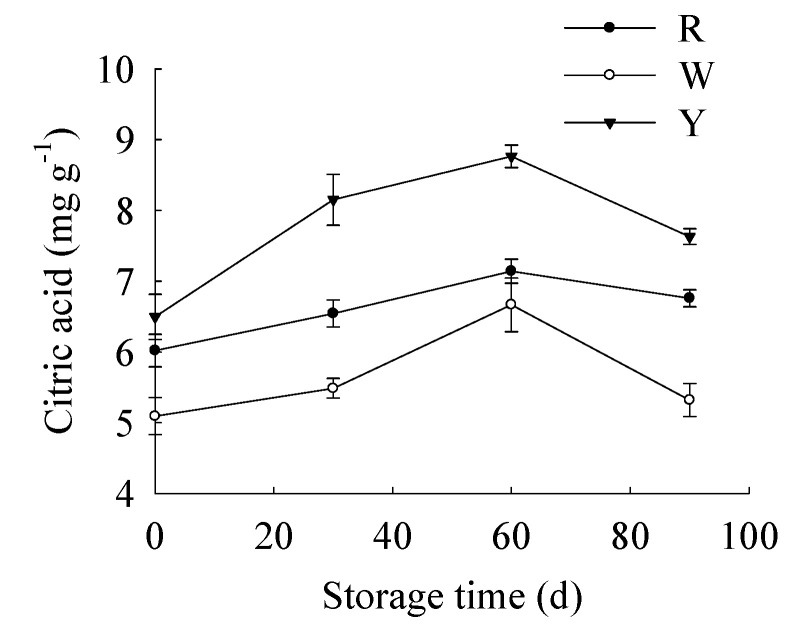
Changes in citric acid content in three pomelo cultivars. Vertical bars represent SE of the means of three replicate assays.

**Figure 7 foods-12-02001-f007:**
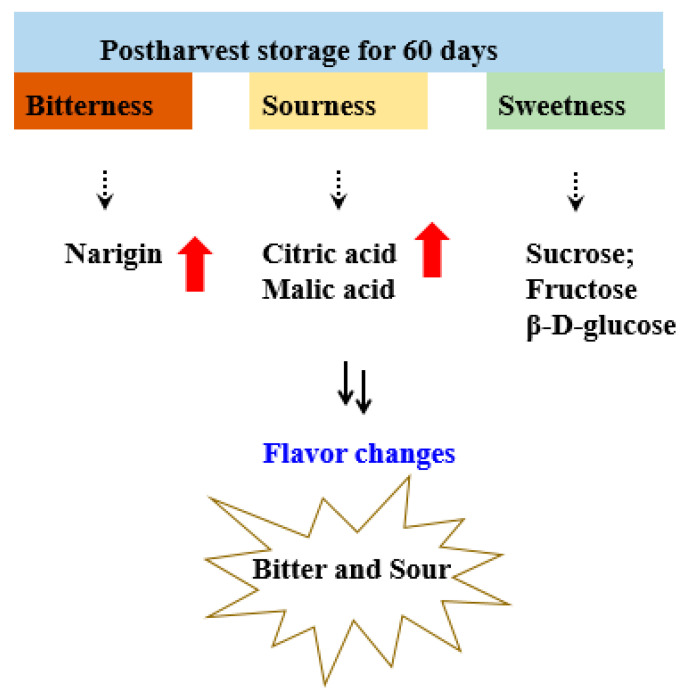
A hypothesis of flavor changes in pomelos during 60 days of storage.

**Table 1 foods-12-02001-t001:** Overview of the metabolites identified in pomelo extracts and assignment of proton signals in the representative ^1^H-NMR spectrum (MeOD-*d*_4_, *δ* in ppm, *J* in Hz).

No.	Metabolites	Assignment of Proton Signals
1	Fatty acids	0.98 (t, *J* = 7.3 Hz), 1.24–1.38 (m), 1.55–1.65 (m), 2.32 (t, *J* = 7.5 Hz)
2	Alanine	1.45 (d, *J* = 7.2 Hz), 3.61 (m)
3	γ-Amino-butyrate (GABA)	1.89 (m), 2.40 (t, *J* = 7.5 Hz), 2.99 (m)
4	Malic acid	2.55 (dd, *J* = 7.7, 16.2 Hz), 2.79 (m), 4.34 (m)
5	Asparagine	2.69 (m), 2.93 (m), 3.82(m)
6	Citric acid	2.73 (d, *J* = 15.4 Hz), 2.81 (d, *J* = 15.4 Hz)
7	Choline	3.20 (s)
8	Phosphocholine	3.22 (s)
9	Unsaturated lipids	5.28–5.33 (m), 2.74–2.80 (m)
10	Fructose	4.09 (d, *J* = 8.3 Hz), 4.01 (m)
11	β-D-Glucose	4.46 (d, *J* = 8.0 Hz), 3.12 (m)
12	α-D-Glucose	5.10 (d, *J* = 3.72 Hz), 3.35 (m)
13	Sucrose	5.38 (d, *J* = 3.8 Hz), 3.42 (m)
14	Phenolics	6.8–8.0 (m)
15	Naringin	2.76 (m), 3.16 (m), 6.15 (d, *J* = 2.20 Hz), 6.17 (d, *J* = 2.20 Hz), 6.81 (d, *J* = 8.20 Hz), 7.1 (d, *J* = 8.20 Hz)

## Data Availability

The data presented in this study are available on request from the corresponding author.

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
