# Peer review of "1H NMR-Based Metabolic Profiling to Follow Changes in Pomelo Cultivars during Postharvest Senescence"

_foods, 2023, doi:10.3390/foods12102001_

Round 1

Reviewer 1 Report

Comments Foods-2258314

The objective of the manuscript ‘1H NMR-based metabolic profiling to follow changes in pumelo cultivars during postharvest senescence’ was to determine metabolites changes in three pumelo cultivars during 90 days of storage. Besides, the hypothesis must be added.

Please indicate the difference between pomelo and pumelo. According to several databases of scientific names, Citrus maxima (Burm.) Merr. is the scientific name of ‘pomelo’. Why do you use ‘pumelo’?

Section 2.1. Indicate the origin’s country of the pomelo cultivars.

Keywords. These could be improved according to the content of the manuscript.  

Section 2.2. Indicate the general conditions of the methods briefly.

Section 2.3 What statistical software did you use? Treatments or the names of the samples should be indicated in another section. PLS-DA model is also a statistical analysis. Why this is in a different section than 2.5.

Section 2 should be reorganized as a logical story. This section must have enough information to replicate the study. Currently, it would not be possible.

The graphical abstract should be improved because the resolution is too low and not according to the manuscript’s findings.

Please read again and carefully throughout the manuscript. It has typography mistakes, inadequate use of capital letters, and some missing blanks. Use subscripts or superscripts appropriately, where appropriate, throughout the document.

Legends of the Figures should be improved. Table and Figures captions and headings must be self-explanatory. Do not forget to indicate the meaning of each acronym, and do not duplicate legends. Besides, names and units in the Figures' axes must always be added. Remember to add the name of the axis, and inside the parentheses, indicate the units.

Indicate the meaning of the acronyms.

The manuscript is well-organized and typed. However, the main concern is regarding Materials and Methods because information for replicate the study is missing. Besides, this section is not well organized.

Insets of Figures should be improved (resolution/quality) to read and identify important information.

This manuscript has several weaknesses regarding the methodology, which is not well described. It is crucial to give additional information to replicate the study and give more detail regarding chemical standards and calibration curves.

The manuscript is ok, but it should be better. However, it requires significant changes regarding the statistical analysis, discussion, and conclusions.

Author Response

Many thanks for the valuable suggestions or comments on our manuscript from the editor and reviewers. As suggested, the manuscript has been examined and revised carefully while the textual and typographical errors have been corrected.

The reviewers’ comments:

Reviewer #1:

The reviewers’ comments:

The objective of the manuscript ‘1H NMR-based metabolic profiling to follow changes in pumelo cultivars during postharvest senescence’ was to determine metabolites changes in three pumelo cultivars during 90 days of storage. Besides, the hypothesis must be added.

The authors’ response:

Thanks for the careful review and valuable comments. Our experiment is to use NMR technology to explore the metabolite changes of different varieties of grapefruit during storage. The research idea is to first use NMR technology to identify the approximate metabolite, and then explore the metabolite changes of pomelo cultivars during storage by comparing the peak area. A hypothesis was added, please check.

The reviewers’ comments:

Please indicate the difference between pomelo and pumelo. According to several databases of scientific names, Citrus maxima (Burm.) Merr. is the scientific name of ‘pomelo’. Why do you use ‘pumelo’?

The authors’ response:

Thanks for the careful review and valuable comments. We have revised it.

The reviewers’ comments:

Section 2.1. Indicate the origin’s country of the pomelo cultivars.

The authors’ response:

Thanks for the careful review and valuable comments. The origin’s country of the pomelo cultivars are from Dapu county, Meizhou city, Guangdong province, China. We have revised it.

The reviewers’ comments:

Keywords. These could be improved according to the content of the manuscript.

The authors’ response:

Thanks for the careful review and valuable comments. We have revised it.

The reviewers’ comments:

Section 2.2. Indicate the general conditions of the methods briefly.

The authors’ response:

Thanks for the careful review and valuable comments. We have revised it.

The reviewers’ comments:

Section 2.3 What statistical software did you use? Treatments or the names of the samples should be indicated in another section. PLS-DA model is also a statistical analysis. Why this is in a different section than 2.5.

The authors’ response:

Thanks for the careful review and valuable comments. We use the statistical software SPSS 22.0 to deal with the other data, but SIMCA 15 (Umetrics, Malmo, Sweden) for PLS-DA analysis. It is quite a good advice, since PLS-DA model is also a statistical analysis, we reorganized the section of statistical analysis. Thanks for the advice.

The reviewers’ comments:

Section 2 should be reorganized as a logical story. This section must have enough information to replicate the study. Currently, it would not be possible.

The authors’ response:

Thanks for the careful review and valuable comments. We have reorganized Section 2 to make sure this part have enough information to replicate the study. Thanks for the advice.

The reviewers’ comments:

The graphical abstract should be improved because the resolution is too low and not according to the manuscript’s findings.

The authors’ response:

Thanks for the careful review and valuable comments. We have revised it.

The reviewers’ comments:

Please read again and carefully throughout the manuscript. It has typography mistakes, inadequate use of capital letters, and some missing blanks. Use subscripts or superscripts appropriately, where appropriate, throughout the document. We have revised it.

The authors’ response:

Thanks for the careful review and valuable comments.

The reviewers’ comments:

Legends of the Figures should be improved. Table and Figures captions and headings must be self-explanatory. Do not forget to indicate the meaning of each acronym, and do not duplicate legends. Besides, names and units in the Figures' axes must always be added. Remember to add the name of the axis, and inside the parentheses, indicate the units.

The authors’ response:

Thanks for the careful review and valuable comments. We have revised it.

The reviewers’ comments:

Indicate the meaning of the acronyms.

The authors’ response:

Thanks for the careful review and valuable comments. We have revised it.

The reviewers’ comments:

The manuscript is well-organized and typed. However, the main concern is regarding Materials and Methods because information for replicate the study is missing. Besides, this section is not well organized.

The authors’ response:

Thanks for the careful review and valuable comments. We have revised it.

The reviewers’ comments:

Insets of Figures should be improved (resolution/quality) to read and identify important information.

The authors’ response:

Thanks for the careful review and valuable comments. We have revised it.

The reviewers’ comments:

This manuscript has several weaknesses regarding the methodology, which is not well described. It is crucial to give additional information to replicate the study and give more detail regarding chemical standards and calibration curves.

The authors’ response:

Thanks for the careful review and valuable comments. Yes, It is crucial to give additional information to replicate the study and give more detail regarding chemical standards and calibration curves, we have revised it, please check.

The reviewers’ comments:

The manuscript is ok, but it should be better. However, it requires significant changes regarding the statistical analysis, discussion, and conclusions.

The authors’ response:

Thanks for the careful review and valuable comments. we have revised the statistical analysis, discussion, and conclusions, please check.

Reviewer 2 Report

The paper in the present form shows many severe pitfalls in the experimentation. Furthermore it is very badly structured and the discussion is very poor.

Here are only some of the several pitfalls:

The H-1 NMR experiments is described in a previous article (ref. 15), but in that paper there is no information about the repetition times used and therefore it is impossible to know if the areas of the signals are subject to saturation phenomena. NMR spectroscopy is known to be a quantitative method if the full relaxation condition is respected: in this case no comparison with other techniques like HPLC is needed, it would have been sufficient to add a known amount of the reference substance (visible in the spectra at 0.0 ppm) to calibrate the concentrations of each metabolite. Otherwise, if the full relaxation condition was not respected, the measured “areas” of the signals are affected by saturation and are not suitable for further analysis.

The evaluation of H-1 spectra are questionable. For instance, the spectrum in fig. 1 shows the two couples of signals of the citric acid, marked as “6”, that are very different while they should be equal: this is due because of the overlap with signals from other metabolites, evident from the analysis of the 2D spectra in fig. 3;

No effort has been done to assign many signals - for instance, those from about 0.7 to about 1.25 ppm have been completely disregarded, although they are evident in the 2D spectra;

Furthermore, the assignment to “phenolics” of the whole area ranging from 6.8 to 8.0 ppm is too generic, and there is no explanation or assignment to those signals

PLS-DA analysis is so poorly described that it hasn't even been reported which is the external variable along which it was performed (time? Cultivar?). No discussion has even been attempted for what has been shown in the score plot, and apparently the only use of the PLS-DA analysis was to find VIP values to identify the important metabolites that were then analyzed by univariate analysis.

In paragraph 2.3 there are 7 lines barely describing the PLS-DA method and 9 lines describing in detail each name given to each sample!

The Discussion includes mostly the description of the behaviour of the selected metabolites in time, that should have been put in the Result section, while the actual discussion is very poor and superficial, like the results related to the total soluble sugars which (possible) fluctuation over time and differences among cultivars is reported but remain unexplained, even with a tentative hypothesis.

NMR-based metabolomics has been widely applied over the last twenty years to the analysis of food and beverage, and in particular on the metabolic profiling to determine varieties, quality and geographical origin, including reviews and book chapters. A much more extensive bibliography should have been reported in this respect.

Author Response

Many thanks for the valuable suggestions or comments on our manuscript from the editor and reviewers. As suggested, the manuscript has been examined and revised carefully while the textual and typographical errors have been corrected.

Reviewer #2:

The reviewers’ comments:

The paper in the present form shows many severe pitfalls in the experimentation. Furthermore it is very badly structured and the discussion is very poor.

The authors’ response:

Thanks for the careful review and valuable comments. We paid more attention to the structure and the discussion.

The reviewers’ comments:

Here are only some of the several pitfalls:

The H-1 NMR experiments is described in a previous article (ref. 15), but in that paper there is no information about the repetition times used and therefore it is impossible to know if the areas of the signals are subject to saturation phenomena. NMR spectroscopy is known to be a quantitative method if the full relaxation condition is respected: in this case no comparison with other techniques like HPLC is needed, it would have been sufficient to add a known amount of the reference substance (visible in the spectra at 0.0 ppm) to calibrate the concentrations of each metabolite.

The authors’ response:

Thanks for the careful review and valuable comments. The repetition time is three in a in our present study and previous article. Our experiment is not strictly quantitative, but semi-quantitative and qualitative. The addition recovery should be done for quantitative analysis, and the relaxation time should be optimized. The main purpose of our study is to show that 1H-NMR metabolic profile can be efficient not only for discrimination of pomelo cultivars, but also for quality control evaluation during the senescence process, therefore, our findings can contribute to the investigation of preservation technology of fruits and vegetables.

The reviewers’ comments:

The evaluation of H-1 spectra are questionable. For instance, the spectrum in fig. 1 shows the two couples of signals of the citric acid, marked as “6”, that are very different while they should be equal: this is due because of the overlap with signals from other metabolites, evident from the analysis of the 2D spectra in fig. 3

The authors’ response:

Thanks for the careful review and valuable comments. This problem involves the difference in the size of the two peaks. The reviewer believes that there are other metabolites superimposed on it, which led him to pull up a bee. There may indeed be compounds superimposed on it, because its hydrogen signal overlaps a lot, but it should not be more important. The more important is that its hydrogen a and hydrogen b may have different states. You can clearly find that, At 2.7, its peak is relatively wide, and at 2.8, its peak is relatively narrow. Therefore, for the size of the two peaks of citric acid, we think that, maybe other metabolites are superimposed inside, but the content of that part is small. The difference in the size of the main peaks may be related to the formation of some carboxyl salts.

The reviewers’ comments:

No effort has been done to assign many signals - for instance, those from about 0.7 to about 1.25 ppm have been completely disregarded, although they are evident in the 2D spectra;

The authors’ response:

Thanks for the careful review and valuable comments. 0.7-1.25ppm is mainly methyl signal of amino acid and organic acid, which is relatively obvious and easy to parse. Because many signals are split and overlapped, and the signal is weak, it is difficult to accurately identify the structure, so many of these signals are not further analyzed. In the next research, we will combine other technologies to quantify metabolites, which can more accurately explain the problem.

The reviewers’ comments:

Furthermore, the assignment to “phenolics” of the whole area ranging from 6.8 to 8.0 ppm is too generic, and there is no explanation or assignment to those signals

The authors’ response:

Thanks for the careful review and valuable comments. Since our experiment is only simple and semi-quantitative, it is difficult to accurately identify the structure of the overlapping part of the signal. The assignment to “phenolics” of the whole area ranging from 6.8 to 8.0 ppm is indeed too generic, in the next study, we will further study the quantitative experiment, and the interpretation or distribution of the signal will be more accurate.

The reviewers’ comments:

PLS-DA analysis is so poorly described that it hasn't even been reported which is the external variable along which it was performed (time? Cultivar?). No discussion has even been attempted for what has been shown in the score plot, and apparently the only use of the PLS-DA analysis was to find VIP values to identify the important metabolites that were then analyzed by univariate analysis.

The authors’ response:

Thanks for the careful review and valuable comments. PLS-DA is a supervised discriminant analysis statistical method. PLS-DA can establish the relationship model between the expression of metabolites and the sample category to realize the prediction of the sample category. Variable Importance for the Projection (VIP) can be calculated to measure the influence intensity and interpretation ability of the expression mode of each metabolite on the classification and discrimination of each group of samples, so as to assist in the screening of marker metabolites (usually with VIP value>1.0 as the screening standard). According to score plot of PLS-DA (Fig. 4A), the first two components (t[1], t[2]) of PLS-DA accounted for 78.9% of the total variance among samples. The samples from three cultivars at 0, 30, 60 and 90 days were both easily discriminated along component 1, indicating that we used PLS-DA is appropriate to analyze the NMR data and further used VIP to screen potential biomarkers. In addition, eight metabolites with VIP exceeding one contributed more to three pomelo cultivars in response to different storage times.

The reviewers’ comments:

In paragraph 2.3 there are 7 lines barely describing the PLS-DA method and 9 lines describing in detail each name given to each sample!

The authors’ response:

Thanks for the careful review and valuable comments. We have revised the section 2, since PLS-DA model is also a statistical analysis, we reorganized the section of statistical analysis. In addition, we moved the name of each sample to the Figure caption. Thanks for the advice.

The reviewers’ comments:

The Discussion includes mostly the description of the behaviour of the selected metabolites in time, that should have been put in the Result section, while the actual discussion is very poor and superficial, like the results related to the total soluble sugars which (possible) fluctuation over time and differences among cultivars is reported but remain unexplained, even with a tentative hypothesis

The authors’ response:

Thanks for the careful review and valuable comments. We have put those descriptions of the behaviour of the selected metabolites in time in the Result section. Indeed the discussion is not well organized, and we have reorganized the discussion part. Please check. We have added a hypothesis in Fig.7, please check.

The reviewers’ comments:

NMR-based metabolomics has been widely applied over the last twenty years to the analysis of food and beverage, and in particular on the metabolic profiling to determine varieties, quality and geographical origin, including reviews and book chapters. A much more extensive bibliography should have been reported in this respect.

The authors’ response:

Thanks for the careful review and valuable comments. We have revised it and cited more references .

Reviewer 3 Report

The aim for this study mentioned by the authors has been achieved: they present the potential of the NMR technology in the metabolomic analysis of fruits (e.g. pumelo), the accuracy of NMR technology compared with other analytical techniques (e.g. HPLC) in determining the metabolites changes during postharvest storage.

Minor recommendations:

Page 2, Line 78:  the authors could explained what’s mean ,,d” e.g. day (d);

Page 11, Line 318: The conclusions can be developed.

Author Response

Reviewer #3:

The reviewers’ comments:

The aim for this study mentioned by the authors has been achieved: they present the potential of the NMR technology in the metabolomic analysis of fruits (e.g. pumelo), the accuracy of NMR technology compared with other analytical techniques (e.g. HPLC) in determining the metabolites changes during postharvest storage.

The authors’ response:

Thanks for the careful review and valuable comments.

The reviewers’ comments:

Page 2, Line 78:  the authors could explained what’s mean ,,d” e.g. day (d);

The authors’ response:

Thanks for the careful review and valuable comments. We have revised it.

The reviewers’ comments:

Page 11, Line 318: The conclusions can be developed.

The authors’ response:

Thanks for the careful review and valuable comments. We have revised it. Please check.

Round 2

Reviewer 1 Report

The authors have adequately addressed my previous comments. Therefore, I think the manuscript could be accepted in its current form.

Author Response

The reviewers’ comments:

The authors have adequately addressed my previous comments. Therefore, I think the manuscript could be accepted in its current form.

The authors’ response:

Many thanks for the valuable suggestions or comments on our manuscript from the reviewers.

Reviewer 2 Report

The revised version does not show any real improvement. Briefly, the main flaws are:

the authors have replied that their NMR experiments are not quantitative, but semi-quantitative and qualitative. This fact, that is not reported in the paper, should have been motivated since most of the NMR-based metabolomics articles published over the last period (more than a hundred) employ NMR spectroscopy as a quantitative method.

Given the 2D spectra shown, a more extensive assignment with more detected metabolites was expected.

The authors make use of PLS-DA analysis in a very rough way, giving a very lacking description of the method: was the external variable - supposingly "time", although it was not stated - used as a quantitative or qualitative one? Do they attempted to make use of PLS-DA with two external variables (time as quantitative and cultivar as qualitative)?

Concerning the results from PLS-DA, the authors claim that “The samples from three cultivars at 0, 30, 60 and 90 days were both easily discriminated along component 1”, while fig. 4A displays that, for example, samples from cultivar Y at time 0 are superimposed to those at times 30 and 90 along axis t[1], providing no discrimination at all along that axis. Furthermore, the authors make use of PLS-DA results only to detect "VIP" metabolites, which behaviors are then analyzed in a classical, univariate way, and no attempt to give any information about the metabolic networks that give rise to the complex variables (t[1] and t[2], in this case) has been taken into account.

The english language is even worsened in the revised manuscript.

Author Response

Many thanks for the valuable suggestions or comments on our manuscript from the reviewers. As suggested, the manuscript has been examined and revised carefully while the textual and typographical errors have been corrected.

Reviewer #2:

The reviewers’ comments:

The revised version does not show any real improvement. Briefly, the main flaws are:

the authors have replied that their NMR experiments are not quantitative, but semi-quantitative and qualitative. This fact, that is not reported in the paper, should have been motivated since most of the NMR-based metabolomics articles published over the last period (more than a hundred) employ NMR spectroscopy as a quantitative method. Given the 2D spectra shown, a more extensive assignment with more detected metabolites was expected.

The authors’ response:

From our experimental perspective, it is to quickly explore the metabolites of pomelo during postharvest aging using NMR technology and compare the quality changes of three varieties of pomelo during storage. We chose a semi quantitative method, which involves directly dissolving the sample with methanol-d4 and conducting a comprehensive preliminary analysis of the metabolites. We did not choose to separate the compounds one by one and compare them with the standard substance, as in quantitative experiments, which may result in missing information in the analysis of 2D spectra. However, the most important thing is that our semi quantitative method has obtained the data we want and preliminarily found the key metabolic substances of pomelo during storage, which is of great significance for quality control of pomelo during postharvest storage. In the following experiments, we will carefully consider the comments made by the reviewers and use quantitative methods to further analyze the changes in flavor quality of pomelo during the postharvest process.

The reviewers’ comments:

The authors make use of PLS-DA analysis in a very rough way, giving a very lacking description of the method: was the external variable - supposingly "time", although it was not stated - used as a quantitative or qualitative one? Do they attempted to make use of PLS-DA with two external variables (time as quantitative and cultivar as qualitative)?

The authors’ response:

Thanks for the careful comments, we have revised the PLS-DA analysis, the external variable "time" is used as quantitative and the external variable "cultivar" is used as qualitative. We made use of PLS-DA with two external variables (time as quantitative and cultivar as qualitative) in our work. This discriminant analysis combined two variables in order to find significant metabolites during storage aging in three pumelo varieties. In the following studies, we will fix one variable and study the role of the other variable in metabolite differentiation.

The reviewers’ comments:

Concerning the results from PLS-DA, the authors claim that “The samples from three cultivars at 0, 30, 60 and 90 days were both easily discriminated along component 1”, while fig. 4A displays that, for example, samples from cultivar Y at time 0 are superimposed to those at times 30 and 90 along axis t[1], providing no discrimination at all along that axis. Furthermore, the authors make use of PLS-DA results only to detect "VIP" metabolites, which behaviors are then analyzed in a classical, univariate way, and no attempt to give any information about the metabolic networks that give rise to the complex variables (t[1] and t[2], in this case) has been taken into account.

The authors’ response:

Thanks for the careful comments, we have revised it. Indeed, samples from cultivar Y at time 0 are superimposed to those at times 30 and 90 along axis t[1], providing no discrimination at all along that axis. We have some misrepresentations. As to the comments on “the authors make use of PLS-DA results only to detect "VIP" metabolites”, we do only use PLS-DA to detect VIP to screen the most important metabolites in three pomelos during postharvest storage, which perfectly fits our research purpose. In the following research, we will make attempt to give any information about the metabolic networks that give rise to the complex variables (t[1] and t[2], in this case) has been taken into account.

The reviewers’ comments:

The english language is even worsened in the revised manuscript.

The authors’ response:

Thanks for the careful review and valuable comments. We have revised it.
